# Portuguese Kelps: Feedstock Assessment for the Food Industry

Diana Pacheco [1] , Giuseppe Miranda [2], Carolina P. Rocha [1], Rosinda L. Pato [3], João Cotas [1] ,
Ana M. M. Gonçalves [1,4] , Sandra M. Dias Santos [3], Kiril Bahcevandziev [3] and Leonel Pereira [1,*]

1 MARE-Marine and Environmental Sciences Centre, Department of Life Sciences, University of Coimbra, 3001-456 Coimbra, Portugal; diana.pacheco@uc.pt (D.P.); carolina.rocha@student.uc.pt (C.P.R.); jcotas@uc.pt (J.C.); amgoncalves@uc.pt (A.M.M.G.)
2 Department of Agriculture, University of Foggia, 71122 Foggia, Italy; giuseppe_miranda.542498@unifg.it
3 Research Centre for Natural Resources Environment and Society (CERNAS), Institute of Applied Research (IIA), Agricultural College of Coimbra (ESAC/IPC), 3045-601 Coimbra, Portugal; rlsp@esac.pt (R.L.P.); sds@esac.pt (S.M.D.S.); kiril@esac.pt (K.B.)
4 Department of Biology and CESAM, University of Aveiro, 3810-193 Aveiro, Portugal
* Correspondence: leonel.pereira@uc.pt; Tel.: +351-239-855-229

**Abstract:** Seaweeds have been incorporated in the daily diet of several human cultures since ancient times, due to their nutritional characteristics and healthy properties. The brown seaweeds *Undaria pinnatifida*, *Saccharina latissima*, *Sacchoriza polyschides*, and *Laminaria ochroleuca* were collected in the Viana do Castelo (Portugal) bay to assess their proximate composition analysis. As a result, the algal biomass was dried, and its moisture and ash content were determined. The dried biomass was then analyzed for total nitrogen/total protein (using the Kjeldahl method), total fiber content (through fiber analyzer digestion), total lipids (in a Soxhlet apparatus), and fatty acid characterization (by gas chromatography-mass spectrometry). Apart from phosphorus, which was analyzed by spectrophotometry, the ashes were employed for mineral and trace element characterization via dry mineralization and quantified using flame atomic absorption spectrometry. Moreover, the total phenolic content was assessed spectrophotometrically by the Folin-Ciocalteu method in the algal aqueous extracts. Analyses showed that their protein concentrations ranged from 12 to 24% dry weight (DW), while lipid concentrations varied between 0.51% and 1.52% DW. Regarding the carbohydrate concentration in these seaweeds, a concentration between 48% and 60% DW was observed. The *S. polyschides* had the highest overall total phenolic content ($6.19 \times 10^{-3}$ g GAE/100 g of dried algae), while *L. ochroleuca* had the lowest amount ($3.72 \times 10^{-3}$ g GAE/100 g of dried algae). *U. pinnatifida* had the highest total fatty acid content (35.13 mg/g DW), whereas *S. latissima* presented the lowest value (22.59 mg/g DW). Significant concentrations of highly unsaturated fatty acids (HUFA) were observed in both seaweeds, with *U. pinnatifida* having the highest value (10.20 mg/g DW) and *S. latissima* the lowest content (4.81 mg/g DW). It is also highlighted that these seaweeds have a nutritional relevance as a source of essential nutrients, including nitrogen, potassium, sodium, calcium, magnesium, and iron.

**Keywords:** brown seaweeds; chemical composition; nutritional value; macro and micronutrients; fatty acids; phenolic compounds

## 1. Introduction

The large brown seaweeds (between 1 and 60 m in length) represent nearly 22% of the world's coastline (biome of 1,469,900 km$^2$) and are responsible for a great part of the flora carbon storage and sequestration. They are commonly denominated as kelps [1–4].

These kelps are cultivated in several countries as a food source, where they play an important role in the food supplementation to prevent the appearance of diseases and illnesses [5–8]. Algal compounds have an impact on the human cell mechanism, promoting benefits or provoking negative reactions in the human welfare [7,9]. Negative impacts in the human welfare are derived by their high mineral content [10,11]. Thus, the most limiting

factor in seaweed direct intake is their high mineral concentration, where the ingestion of 5 g/day of dried seaweed biomass is the recommended dosage of some minerals, enough to provide several human health benefits [7,12].

Furthermore, it is necessary to study the seaweed's nutritional value and the parameters that affect its quality, such as seawater quality and seaweed nutritional variation in different locations where seaweeds are found, in order to evaluate the seaweed's potential to be incorporated as a direct food source [13].

In Portugal, four species exist (Figure 1a–d) and are considered kelps: the native species *Laminaria ochroleuca*, *Saccharina latissima*, and *Saccorhiza polyschides* and the non-native species *Undaria pinnatifida* [13–15]. *Laminaria ochroleuca* grows in extensive populations in deep intertidal pools and relatively sparse patches in the upper subtidal zone in the northern region of Portugal, whereas *Saccorhiza polyschides* dominates the subtidal zone [16,17].

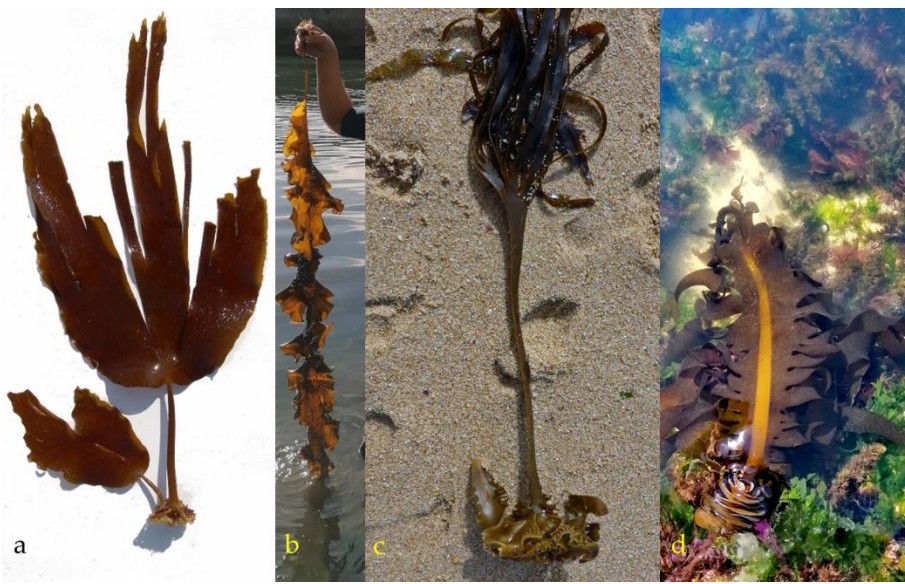

**Figure 1.** Photographic record of (**a**) Laminaria ochroleuca, (**b**) Saccharina latissima, (**c**) Saccorhiza polyschides, and (**d**) Undaria pinnatifida.

Between 1999 and 2007, *Undaria pinnatifida* was only found in two locations in the north of Portugal; therefore, it was thought to be a recent introduction that has begun to colonize intertidal coastlines [18,19].

These seaweeds are more localized in the northern area, mainly in the Viana do Castelo region, where *L. ochroleuca* and *S. latissima* are considered stable populations [14,20]. The presence of *S. polyschides* and the non-native species *U. pinnatifida* appears to replace the ecological niche of *L. ochroleuca* and *S. latissima* in the southern regions, due to their capacity to survive in warmer seawater and better adaptability to environmental changes [13,14].

Kelps are used mainly as a raw food source or alginate extraction for the food industry [21–23]. As raw food, *U. pinnatifida* (known as wakame) is one of the most consumed seaweeds in the world, mainly in the Asiatic cuisine, and is regularly used as a food supplement in several cooked dishes, like *L. ochroleuca* (known as kombu) and *S. latissima* (known as sugar kelp) [22–24]. In the food industry, alginate extracted from these seaweeds is usually applied as a food additive (mainly *L. ochroleuca* and *S. latissima*) due to its inherent ability to stabilize, emulsify liquids, and to form gels of liquids. Thus, they are normally applied in jams, jellies, ice creams, and water-oil solutions (such as mayonnaise). In this case, they can also be added, in mild acid processed foods, as stabilizers [21]. In current times, the potential of the seaweeds *U. pinnatifida* and *S. latissima* is being studied towards the quest of novel ingredients in the food industry, such as fucoidan (polysaccharide), phlorotannin's (phenolic compounds), and fucoxanthin (pigments) [5,25]. However, the

seaweed *S. polyschides* is not so much exploited by the food industry but it is the basis of "Sargaço" (a natural seaweed mixture used traditionally in agriculture in Portugal [26–29]).

The seaweeds *L. ochroleuca* and *S. latissima* have a high biomass productivity due to their large size, and it is possible for them to be cultivated, whereas *S. polyschides* and *U. pinnatifida* have a fast growth rate [13,14,30,31]. Despite the information gathered and the studies done with these seaweed species, there is a general lack of the nutritional and mineral profile of these species, which can be a key for global food safety [5]. The goal of this research is to determine the nutritional value of four Portuguese kelps (*L. ochroleuca*, *S. latissima*, *S. polyschides*, and *U. pinnatifida*), which are found in two co-habiting locations in the Viana do Castelo region (northern Portugal), as well as to determine their nutritional potential and limiting components for human consumption, in order to monitor the potential exploitation of these species in the functional food supplements sector with the goal of promoting a healthier diet and nutraceutical foods, which is critical given the rise in the prevalence of food-related disorders like diabetes and obesity [32].

## 2. Materials and Methods

### 2.1. Seaweed Harvesting and Preparation

During the summer of 2020 (24th of July), the brown seaweeds were collected from two sites of Viana do Castelo. The seaweeds *Saccorhiza polyschides* and *Laminaria ochroleuca* were collected in Praia da Amorosa (41°38′23.4″ N 8°49′19.6″ W), whereas *Saccharina latissima* and *Undaria pinnatifida* were harvested in the entrance of the Viana do Castelo harbor (41°41′17.7″ N 8°50′11.4″ W). The online database AlgaeBase was used to categorize these seaweeds [33].

Afterwards, seaweeds were transported in plastic bags in a cool box to the laboratory and frozen at −20 °C for prior utilization. Some days later, the seaweeds were washed with filtered seawater (collected in the sampling sites) to remove the sand, epiphytes, and other detritus. Then, the biomass was washed with distilled water to remove the salt content of seawater, placed in plastic trays, and dried in an air-forced oven (Raypa DAF-135, R. Espinar S.L., Barcelona, Spain) for 48 h at 60 °C. After this procedure, the biological samples were milled (<1 cm) with a commercial grinder (Taurus Aromatic, Oliana, Spain) and stored in sterile flasks in a dark and dry place at room temperature.

### 2.2. Moisture and Ashes Content

According to the international standard method 930.04 of Official Methods of Analysis of AOAC International [34], the moisture content was assessed through the fresh weight of the algal samples after oven drying (Memmert, Büchenbach, Germany) at 60 °C for 48 h. Afterwards, the samples were milled (<1 mm), and approximately 2 g of each sample was placed in crucibles and dried at 105 °C for 2 h. Then, the samples were placed in a desiccator, being again weighed (Mettler Toledo, Columbus, OH, USA) to calculate the moisture content. In accordance with the AOAC method 930.05, the samples dried at 105 °C were placed in an incineration muffle for 2 h at 550 °C (Induzir, Portugal) and further cooled in a desiccator and weighed (Mettler Toledo, Columbus, OH, USA) to assess the ashes amount.

The moisture at 65 °C was calculated according to standard method 930.04 of AOAC [34]:

$$Moisture\ at\ 60°C(\%) = \frac{(P2 - P3)}{(P2 - P1)} \times 100$$

*P*1—weight of the tray (g); *P*2—weight of the tray + sample (g); *P*3—weight of the tray + dried sample (g).

The moisture at 65 °C to 105 °C was calculated according to standard method 930.04 of AOAC [34]:

$$Moisture\ (60°C - 105°C)\ (\%) = \frac{(P5 - P6)}{(P5 - P4)} \times 100$$

*P4*—crucible weight (g); *P5*—crucible weight + sample (g); *P6*—crucible weight + dried sample (g).

The moisture content was calculated according to standard method 930.04 of AOAC [34]:

$$Moisture\ (\%) = \frac{\frac{(P5-P4)\times(P2-P1)}{(P3-P1)-(P6-P4)}}{\frac{(P5-P4)\times(P2-P1)}{P3-P1}} \times 100$$

The ashes content was calculated according to standard method 930.05 of AOAC [34]:

$$Ashes\ (\%\ db) = 100 \times \frac{(P5-P6)}{(P5-P4)}\ Ashes\ (\%\ fb) = \frac{ashes\ (\%\ db) \times (100-H)}{100}$$

% *db*—percentage of dried biomass; % *fb*—percentage of fresh biomass; *P4*—crucible weight (g); *P5*—crucible weight + sample (g); *P6*—crucible weight + ashes (g); *H*—moisture (%).

### 2.3. Crude Lipids

The total lipids content was gravimetrically quantified following a continuous extraction process with diethyl ether in a Soxhlet apparatus (Behr Labor-Technik GmbH, Düsseldorf, Germany), as it follows the international standard AOAC method 930.09 [34]. The distillation flasks were previously dried at 105 °C for 2 h, cooled in a desiccator, and weighed using an analytical scale (Sartorix, Göttingen, Germany). Afterwards, the distillation flasks were filled (to 2/3 of their capacity) with diethyl ether (Panreac, Darmstadt, Germany). Then, approximately 2 g of the algal samples (Sartorix, Göttingen, Germany) were packed in filter paper and placed into the thimble. After 16 h of extraction, all the solvent was collected and evaporated (BÜCHI Labortechnik AG, Flawil, Switzerland). The distillation flasks were then dried at 105 °C for 2 h and weighed (Sartorix, Germany) when cooled down.

Crude lipids were calculated according to the formula presented by the standard method of AOAC 930.09 [34]:

$$Crude\ lipids\ (\%\ db) = 100 \times \frac{P3-P1}{P2}$$

$$Crude\ lipids\ (\%\ fb) = \frac{Crude\ lipids\ (\%\ db) \times (100-H)}{100}$$

% *db*—percentage of dried biomass; % *fb*—percentage of fresh biomass; *P1*—distillation flask weight (g); *P2*—sample weight (g); *P3*—distillation flask weight + lipids (g); *H*—moisture (%).

Fatty Acid Analysis

Fatty acids were extracted from dry algal biomass and trans-methylated to fatty acid methyl esters (FAMEs) for analysis as described by Gonçalves et al. (2012) [35]. Samples were incubated with methanol (Fisher Chemical, Waltham, MA, USA) for the extraction of lipids. The nonadecanoic acid (C19:0) (Fluka 74208) was added as an internal standard for further quantification. Samples were centrifuged (Sorvall™ ST16, Thermo Scientific, Waltham, MA, USA) for 25 min at 5 °C, 537 g, and stored in liquid form at −80 °C until further analysis.

FAMEs' identification was done by gas chromatography-mass spectrometry (GC-MS) using Thermo Scientific Trace 1310 Network (Waltham, MA, USA) equipment, equipped with a TR-FFAP column (Thermo Scientific, Waltham, MA, USA) of 0.32 mm internal diameter, 0.25 µm film thickness, and 30 m long. The sample (0.60 µL) was injected in splitless mode, at an injector temperature of 250 °C, lined with a split glass liner of 4.0 mm i.d. The initial oven temperature was 80 °C, following a linear temperature increase of 25 °C min$^{-1}$ to 160 °C, followed by another ramp of 2 °C min$^{-1}$ to 210 °C, and finally an increase of 40 °C min$^{-1}$ until a final temperature of 230 °C was reached and maintained for 10 min. Helium at a flow rate of 1.4 mL min$^{-1}$ was used as a carrier gas. A Thermo Scientific ISQ 7000 Network Mass Selective Detector (Waltham, MA, USA) at scanning m/z

ranges specific for fatty acids in Selected Ion Monitoring (SIM) mode acquisition was used. The detector starts operating 3.5 min after injection, which corresponds to solvent delay. The injector ion source and transfer line were maintained at 240 °C and 230 °C, respectively. Integration of FAME peaks were carried out using the equipment's software Xcalibur ™ (Thermo Scientific, Waltham, MA, USA). Identification of each peak was performed by retention time and mass spectrum of each FAME, comparing to the Supelco® 37 component FAME mix (Sigma-Aldrich, Steinheim, Germany). Quantification of FAMEs was done as described in Gonçalves et al. (2012) [35].

### 2.4. Total Nitrogen/Protein

The total nitrogen/protein content was determined by the Kjeldahl method (AOAC method 978.04) [35], whilst 5 was used as a protein conversion factor [36]. Approximately 0.5 g (Mettler Toledo, Columbus, OH, USA) of the previously dried algal sample was added to a Kjeldahl tube, and then a selenium catalyst (PanReac AppliChem, Darmstadt, Germany) and 12 mL of sulfuric acid (Chem-Lab NV, Zedelgem, Belgium) was added. The tubes were then placed into the Kjeldahl digester (VELP Scientifica, Usmate Velate MB, Italy) at 400 °C for 2 h. The samples were allowed to cool in the fume cupboard, and 50 mL of distilled water was added to each tube and put into the Kjeldahl distiller (VELP Scientifica, Usmate Velate MB, Italy). Concurrently, 30 mL of boric acid (Chem-Lab NV, Zedelgem, Belgium) was placed in an Erlenmeyer (one for each sample), being further placed into the Kjeldahl distiller as well (VELP Scientifica, Usmate Velate MB, Italy). To the Kjeldahl tube, 50 mL of distilled water and 50 mL of sodium hydroxide (NaOH) at 40% (*m/v*) (JMGS—José Manuel Gomes dos Santos, Portugal) were added. The distilled solution was collected and titrated with chloridric acid (HCl) 0.1 M (Chem-Lab NV, Zedelgem, Belgium).

Total protein was calculated according to the formula [34]:

$$Total\ protein\ (\%\ db) = fator \times 100 \times \frac{0.01401 \times [HCl] \times (V-V0)}{P1 \times 10}$$

$$Total\ protein\ (\%\ fb) = \frac{Total\ protein\ (\%\ db) \times (100-H)}{100}$$

% *db*—percentage of dried biomass; % *fb*—percentage of fresh biomass; $P1$—sample weight (g); $[HCl]$—chloridric acid concentration (M); $V$—volume of titrant spent in sample titration (mL); $V0$—volume of titrant spent in control sample titration (mL); $H$—moisture (%).

### 2.5. Crude Fiber and Total Carbohydrates/Nitrogen-Free Extractives

According to the standard method 930.10 of AOAC [34], the crude fiber was analyzed through the weighing of 2 g (Sartorix, Göttingen, Germany) from the algal samples, previously oven dried (Memmert, Büchenbach, Germany) at 105 °C for 2 h and placed in a 600 mL goblet. It was then added 200 mL of sulfuric acid ($H_2SO_4$) 12.5 g/L (Chem-Lab NV, Zedelgem, Belgium), and the samples were placed in a fiber analyzer (Labconco Corporation, Kansas City, MO, USA) for 30 min. After this procedure, the samples were filtered with a filter crucible G2 under vacuum (General Electric, Boston, MA, USA). The residue was then placed into the goblet with 250 mL of sodium hydroxide (NaOH) 12.5 g/L (JMGS—José Manuel Gomes dos Santos, Odivelas, Portugal) and set into the fiber analyzer (Labconco Corporation, Kansas City, MO, USA) for an additional 30 min. With the same filter crucible G2 (Robu, Hattert, Germany), the samples were again vacuum filtered and dried at 130 °C for 2 h. After the samples were cooled down in a desiccator, they were weighed using an analytical scale (Sartorix, Göttingen, Germany) and placed into an incineration muffle at 550 °C (Induzir, Batalha, Portugal) for 2 h. Finally, the samples could cool down and were weighed (Sartorix, Göttingen, Germany) to calculate the crude fiber. Nitrogen-free extractives are the difference for 100 of the remaining constituents (moisture, lipids, protein, crude fiber, and ash), while the total carbohydrates correspond

approximately to the difference between 100 and the sum of the moisture, ash, lipids, and protein.

Total fiber was calculated according to the formula [37]:

$$Crude\ fiber\ (\%\ db) = 100 \times \frac{P2 - P3}{P1}$$

$$Crude\ fiber\ (\%\ fb) = \frac{Crude\ fiber\ (\%\ db) \times (100 - H)}{100}$$

% *db*—percentage of dried biomass; % *fb*—percentage of fresh biomass; *P1*—sample weight (g); *P2*—crucible weight + sample dried at 130 °C (g); *P3*—crucible weight + sample dried at 550 °C (g); *H*—moisture (%).

### 2.6. Mineral and Trace Element Characterization

With the ashes obtained, the mineral content was analyzed through dry mineralization and assessed by using flame atomic absorption spectrometry (PerkinElmer PinAAcle 900 T, Waltham, MA, USA) [38]. Apart from phosphorus analysis that was performed by spectrophotometry (PG instruments T80+ UV/VIS spectrophotometer, Leicestershire, United Kingdom) [39].

For this analysis, it was performed an acid digestion with nitric acid 65% (*m/v*) (Chem-Lab NV, Zedelgem, Belgium), in a water bath at 100 °C for about 30 min. Finally, the samples were filtrated for a volumetric flask and the volume adjusted with distilled water. After the necessary dilutions (1:10, 1:100 and 1:500), the analysis was carried out on the atomic absorption spectrophotometer equipped with the cathode corresponding to each element.

### 2.7. Total Phenolic Compounds Quantification

Seaweeds were dried at 40 °C for 48 h; then a crude extract was conducted using distilled water (10:100 m:v) in a Moulinex LM811D11 blender (SEB, Selongey, France). After the liquification of seaweeds, the crude extracts were filtered with a Buchner and Goosh (Linex, Barcelona, Spain) filter (G2 porosity) under vacuum [40].

The phenolic compounds were quantified using the Folin-Ciocalteu method, and gallic acid was used as the standard units (GAE—gallic acid equivalent units). For the analysis, 450 μL of crude extract, 50 μL of Folin-Ciocalteu reagent (Panreac, Barcelona, Spain), 1000 μL of aqueous solution of sodium carbonate (Chem-Lab NV, Zedelgem, Belgium) (75 g/L m:v), and 1000 μL of distilled water were added to the tubes. The samples were immediately vortexed for 30 s and incubated in the dark for 30 min at room temperature. The absorbance of the supernatant was measured at 750 nm using a Hitachi 2000 (Hitachi 2000, Tokyo, Japan).

To quantify the total phenolic content, a standard curve was performed (y = 0.0168x + 0.0159; $r^2$ = 0.9998) with different concentration of gallic acid (0, 4, 6, 8, 10, 20, 40, 60 mg GAE/L).

### 2.8. Statistical Analysis

The statistical analysis was performed with the software Sigma Plot v.14. Data was checked for normality (Shapiro–Wilk test) and homogeneity (the equal variance test Brown-Forsythe). One-way analysis of variance (ANOVA) was then performed to assess statistically significant differences between each nutritional parameter within the algal samples. The statistical analysis was performed comparing the different algal samples, being considered statistically different when *p* value < 0.05. The Tukey multiple comparison t-test was used after the rejection of the one-way ANOVA null hypothesis (Tukey Test).

The fatty acid profile was statistically analyzed and compared the variation in FA composition through non-metric multidimensional scaling (n-MDS). One-way analysis of similarity (ANOSIM) was applied to test differences in FA profiles across seaweed species. Similarity percentage analysis (SIMPER) was performed to test the contribution of individual FA to similarities and dissimilarities within and between sample groups.

Analysis of variance (ANOVA) was also conducted to assess differences in the studied components between seaweed species.

The results obtained through the nutritional, mineral, and trace element characterization were determined in duplicate, being the results presented by mean ± standard error and expressed by g 100 g$^{-1}$. The fatty acids analysis was determined in triplicate, being the results presented by mean ± standard deviation and expressed by mg g$^{-1}$. Finally, the total phenolic content (TPC) in the crude aqueous extracts was determined in triplicate, being the results presented by mean ± standard deviation and expressed by g GAE/100 g of algae (dry weight basis).

## 3. Results

All the seaweeds analyzed demonstrated a different micronutrient and trace element profile (Table 1), containing different amounts of each element. For instance, the non-indigenous seaweed *U. pinnatifida* exhibited the highest content of nitrogen, phosphorus, and sodium (3.8 g N 100 g$^{-1}$, 0.223 g P 100 g$^{-1}$, and 1.40454 g Na 100 g$^{-1}$, respectively), while containing the lowest amount of potassium, iron, copper, and manganese (0.55303 g K 100 g$^{-1}$, 0.00682 g Fe 100 g$^{-1}$, 0.0002 g Cu 100 g$^{-1}$, and 0.00038 g Mn 100 g$^{-1}$, respectively). However, the brown seaweed *S. polyschides* demonstrated its high abundance of calcium and magnesium (0.68929 g Ca 100 g$^{-1}$ and 0.44725 g Mg 100 g$^{-1}$). In comparison with the other analyzed seaweeds, *S. latissima* exhibited the highest concentration of iron (0.06265 g Fe 100 g$^{-1}$), copper (0.00053 g Cu 100 g$^{-1}$), and zinc (0.00285 g Zn 100 g$^{-1}$); in contrast, it presented the lowest amount of phosphorus (0.111048 g P 100 g$^{-1}$) and calcium (0.44427 g Ca 100 g$^{-1}$). For another perspective, *L. ochroleuca* showed to be the richest source of manganese (0.00126 g Mn 100 g$^{-1}$) but the poorest of nitrogen (2.15 g N 100 g$^{-1}$), magnesium (0.26609.g Mg 100 g$^{-1}$), sodium (1.07026 g Na 100 g$^{-1}$), and zinc (0.0019 g Zn 100 g$^{-1}$).

**Table 1.** Micronutrient and trace element composition of *U. pinnatifida*, *S. latissima*, *L. ochroleuca*, and *S. polyschides*. Results are expressed in mean ± standard error (*n* = 2; dry weight basis). Statistically significant differences in the same element content among the species are expressed by different letters.

| Micronutrient and Trace Element (g 100 g$^{-1}$) | *U. pinnatifida* | *S. polyschides* | *S. latissima* | *L. ochroleuca* |
|---|---|---|---|---|
| Nitrogen | 3.8 ± 0.02160 [a] | 2.37 ± 0.08718 [b] | 2.31 ± 0.03877 [b] | 2.15 ± 0.02770 [b] |
| Phosphorus | 0.223 ± 0.00059 [a] | 0.11705 ± 0.00930 [b] | 0.11048 ± 0.00026 [b] | 0.1319 ± 0.00724 [b] |
| Calcium | 0.60928 ± 0.03119 [a] | 0.68929 ± 0.01117 [a] | 0.44427 ± 0.01368 [b] | 0.65807 ± 0.01312 [a] |
| Magnesium | 0.42665 ± 0.00778 [a] | 0.44725 ± 0.00374 [a] | 0.28286 ± 0.00014 [b] | 0.26609 ± 0.00027 [b] |
| Potassium | 0.55303 ± 0.07497 [a] | 1.54396 ± 0.00890 [b] | 0.25134 ± 0.01345 [c] | 1.23641 ± 0.00131 [d] |
| Sodium | 1.40454 ± 0.07540 | 1.09421 ± 0.00747 | 1.2572 ± 0.02229 | 1.07026 ± 0.05769 |
| Iron | 0.00682 ± 0.29554 [a] | 0.01605 ± 0.72364 [b] | 0.06265 ± 0.13051 [c] | 0.0083 ± 0.18507 [d] |
| Copper | 0.0002 ± 0.04408 | 0.00023 ± 0.17871 | 0.00053 ± 0.42157 | 0.00031 ± 0.80396 |
| Zinc | 0.00277 ± 0.07034 [a] | 0.00262 ± 0.62707 [a] | 0.00285 ± 0.12177 [a] | 0.0019 ± 0.26826 [b] |
| Manganese | 0.00038 ± 0.23138 [a] | 0.00046 ± 0.50164 [a] | 0.00045 ± 0.90370 [a] | 0.00126 ± 1.08028 [b] |

[a,b,c,d] Similar letters indicate no significant differences at the *p*-value < 0.05 level.

Fresh (Table 2) and dry (Table 3) seaweed biomass hold different nutritional characteristics. The fresh biomass of *U. pinnatifida* showed a different nutritional characterization, being the poorest seaweed in ash (1.20 g 100 g$^{-1}$), fat (0.04 g 100 g$^{-1}$), fiber (0.57 g 100 g$^{-1}$), protein (1.53 g 100 g$^{-1}$), carbohydrates (3.14 g 100 g$^{-1}$), and energy value (80 KJ/100 g). Conversely, *S. latissima* was the nutritionally richest seaweed, presenting the highest values of fat (0.21 g 100 g$^{-1}$), protein (1.90 g 100 g$^{-1}$), carbohydrates (8.45 g 100 g$^{-1}$), and energy (181 KJ/100 g). While *U. pinnatifida* exhibited the highest moisture (93.5 g 100 g), *L. ochroleuca* presented the lowest (85.9 g 100 g$^{-1}$), offering the highest carbohydrate content, with 1.64 g 100 g$^{-1}$. Overall, *U. pinnatifida* is the seaweed with the lowest nutritional content among the seaweeds investigated.

**Table 2.** Nutritional characterization of the fresh biomass of *U. pinnatifida*, *S. latissima*, *L. ochroleuca*, and *S. polyschides*. Results are expressed in mean ± standard error. Statistically significant differences in the same element content among the species are expressed by different letters. (*n* = 2; fresh weight basis).

| | *U. pinnatifida* | *S. polyschides* | *S. latissima* | *L. ochroleuca* |
|---|---|---|---|---|
| Moisture (g 100 g$^{-1}$) | 93.52 ± 0.007 [a] | 87.09 ± 0.028 [b] | 86.07 ± 0.007 [c] | 85.9 ± 0.007 [d] |
| Ash (g 100 g$^{-1}$) | 1.20 ± 0.007 [a] | 2.70 ± 0.007 [b] | 2.48 ± 0.007 [c] | 2.58 ± 0.007 [d] |
| Fat (g 100 g$^{-1}$) | 0.04 ± 0.007 [a] | 0.19 ± 0.007 [b] | 0.21 ± 0.014 [b] | 0.07 ± 0.007 [a] |
| Fiber (g 100 g$^{-1}$) | 0.57 ± 0.007 [a] | 1.36 ± 0.014 [b] | 0.89 ± 0.014 [c] | 1.64 ± 0.028 [d] |
| Protein (g 100 g$^{-1}$) | 1.53 ± 0.014 [a] | 1.81 ± 0.007 [b] | 1.90 ± 0.007 [b] | 1.81 ± 0.035 [b] |
| Nitrogen-Free Extractives (g 100 g$^{-1}$) | 3.14 ±0.007 [a] | 6.85 ± 0.014 [b] | 8.45 ± 0.014 [c] | 7.99 ± 0.014 [d] |
| Energy (Kcal/100 g) | 19.00 ± 0.028 [a] | 36.00 ± 0.134 [b] | 43.00 ± 0.177 [c] | 40 ± 0.120 [d] |
| Energy (KJ/100 g) | 80.00 ± 0.106 [a] | 152.00 ± 0.566 [b] | 181.00 ± 0.757 [c] | 167 ± 0.516 [d] |

[a,b,c,d] Similar letters indicate no significant differences at the *p*-value < 0.05 level.

**Table 3.** Nutritional characterization of the dried biomass of *U. pinnatifida*, *S. latissima*, *L. ochroleuca*, and *S. polyschides*. Results are expressed in mean ± standard error (*n* = 2; dry weight basis). Statistically significant differences in the same element content among the species are expressed by different letters.

| | *U. pinnatifida* | *S. polyschides* | *S. latissima* | *L. ochroleuca* |
|---|---|---|---|---|
| Ash (g 100 g$^{-1}$) | 18.58 ± 0.049 [a] | 20.89 ± 0.035 [b] | 17.81 ± 0.049 [c] | 18.33 ± 0.07 [d] |
| Fat (g 100 g$^{-1}$) | 0.63 ± 0.099 [a] | 1.50 ± 0.042 [b] | 1.52 ± 0.106 [b] | 0.51 ± 0.049 [a] |
| Fiber (g 100 g$^{-1}$) | 8.8 ± 0.085 [a] | 10.54 ± 0.092 [b] | 6.39 ± 0.127 [c] | 11.65 ± 0.212 [d] |
| Protein (g 100 g$^{-1}$) | 23.54 ± 0.219 [a] | 14.01 ± 0.064 [b] | 13.63 ± 0.021 [c,d] | 12.83 ± 0.247 [d] |
| Nitrogen-Free Extractives (g 100 g$^{-1}$) | 48.44 ± 0.007 [a] | 53.05 ± 0.057 [b] | 60.64 ± 0.092 [c] | 56.68 ± 0.028 [d] |
| Energy (Kcal/100 g) | 294 ± 0.057 [a] | 282 ± 0.431 [b] | 311 ± 1.223 [c] | 283 ± 0.643 [b] |
| Energy (KJ/100 g) | 1229 ± 0.226 [a] | 1180 ± 1.810 [b] | 1301 ± 5.127 [c] | 1183 ± 2.694 [b] |

[a,b,c,d] Similar letters indicate no significant differences at the *p*-value < 0.05 level.

Regarding the ash content, *S. polyschides* exhibits the highest amount (20.89 g 100 g$^{-1}$), while *S. latissima* the lowest amount (17.81 g 100 g$^{-1}$). The non-indigenous seaweed species *U. pinnatifida* showed the lowest amount of nitrogen-free extractives (or approximately the carbohydrate content), presenting 48.44 g 100 g$^{-1}$, but demonstrated to be a rich protein source (23.54 g 100 g$^{-1}$). *S. polyschides* and *L. ochroleuca* stand out for the lowest energy value, where a 100 g portion provides 1180 and 1183 KJ, respectively. Compared with the other analyzed seaweeds, *S. latissima* stands out positively for having the highest amount of carbohydrates or nitrogen-free extractives (60.64 g 100 g$^{-1}$) and for providing the highest energy value (1301 KJ/100 g). In contrast, this seaweed has the lowest fiber content, presenting 6.39 g 100 g$^{-1}$. Furthermore, the species *L. ochroleuca* exhibited the highest concentration of fiber (11.65 g 100 g$^{-1}$) and the lowest content of fat and protein (0.51 and 12.83 g 100 g$^{-1}$, respectively). Moreover, the seaweeds *S. latissima* and *S. polyschides* also stand out positively for their high total lipid content (1.52 and 1.50 g 100 g$^{-1}$, respectively).

Fatty acid analysis allowed the identification of saturated fatty acids (SFA), monounsaturated fatty acids (MUFA), polyunsaturated fatty acids (PUFA), and highly unsaturated fatty acids (HUFA) in the studied species, with a particular interest in the omega-3 fatty acids encountered (Table 4). In terms of total fatty acids per gram of dried algae, *Undaria pinnatifida* was the species presenting the highest value, with the contribution of HUFA being particularly high (29% of total FA). Saturated fatty acids were the most abundant class of FA with C16:0 dominating in all species among all fatty acid classes.

Significant differences between the four species studied regarding their FA content were somewhat masked by the standard deviations among replicates (Table 4). Nonetheless, clear differences can be observed regarding the contents of C18:3c, C18:3t, and EPA, whereas *Sacchorizha polyschides* and *Undaria pinnatifida* clearly present higher content of the mentioned FA compared to *Laminaria ochroleuca* and *Saccharina latissima*.

**Table 4.** Fatty acid (FA) profile and content of each fatty acid (expressed in mg g$^{-1}$) of the four algae species studied. SFA—saturated fatty acids; MUFA—monounsaturated fatty acids; PUFA—polyunsaturated fatty acids; HUFA—highly unsaturated fatty acids; LA—linolenic acid. N corresponds to the number of different fatty acids found for each species. Results are expressed in mean $\pm$ standard deviation (*n* = 3). Statistically significant differences in the same fatty acid content among the species are expressed by letters above the bars ([a,b,c,d]). Similar letters indicate no significant differences at the *p*-value < 0.05 level.

| FA | *L. ochroleuca* | | | *S. latissima* | | | *S. polyschides* | | | *U. pinnatifida* | | |
|---|---|---|---|---|---|---|---|---|---|---|---|---|
| C14:0 | 1.12 [a] | ± | 0.06 | 2.22 [b] | ± | 0.48 | 1.53 [b] | ± | 0.04 | 1.90 [b] | ± | 0.22 |
| C16:0 | 6.73 [a] | ± | 0.48 | 5.73 [a] | ± | 1.22 | 7.26 [a] | ± | 0.23 | 8.56 [b] | ± | 0.87 |
| Σ SFA | | **7.85** | | | **7.80** | | | **8.80** | | | **10.46** | |
| C16:1 | 1.12 [a] | ± | 0.05 | 1.52 [a] | ± | 0.52 | 2.02 [b] | ± | 0.01 | 1.27 [a] | ± | 0.10 |
| C18:1 | 4.39 [a] | ± | 0.09 | 5.53 [a] | ± | 1.01 | 3.47 [a] | ± | 0.58 | 4.37 [a] | ± | 0.42 |
| Σ MUFA | | **5.51** | | | **7.05** | | | **5.49** | | | **5.64** | |
| C18:3 (α-LA) | 0.74 [a] | ± | 0.05 | 0.42 [a] | ± | 0.10 | 1.83 [b] | ± | 0.06 | 2.16 [b] | ± | 0.13 |
| C18:3 (γ-LA) | 1.26 [a] | ± | 0.12 | 0.57 [b] | ± | 0.14 | 2.96 [c] | ± | 0.11 | 2.97 [c] | ± | 0.37 |
| Σ PUFA | | **2.00** | | | **0.99** | | | **4.79** | | | **5.13** | |
| 20:5 (ω-3) | 2.03 [a] | ± | 0.07 | 1.66 [a] | ± | 0.28 | 4.64 [b] | ± | 0.32 | 4.29 [b] | ± | 0.53 |
| 22:6 (ω-3) | 1.39 [a] | ± | 0.05 | 0.98 [a] | ± | 0.22 | 1.06 [a] | ± | 0.67 | 1.54 [a] | ± | 0.30 |
| Σ HUFA | | **3.42** | | | **2.64** | | | **5.70** | | | **5.82** | |
| Σ FA | | **18.78** | | | **18.48** | | | **24.77** | | | **27.05** | |
| ω-6/ω-3 | | **0.30** | | | **0.19** | | | **0.39** | | | **0.37** | |
| N | | 8 | | | 8 | | | 8 | | | 8 | |

Regarding the results, there are three different groups (Figure 2): the first composed of *Laminaria ochroleuca* replicates (A), the second of *Saccharina latissima* replicates (B), and the third of *Sacchorizha polyschides* and *Undaria pinnatifida* replicates (C). There are also two replicates that may be considered as outliers (SL_R2 and SP_R3). The analysis shows a significant difference between groups A and C and between groups B and C. Regarding the differences between the groups A and C, EPA, γ-LA, and α-LA were, in this order, the main FAs contributing to the differences between the groups. Regarding groups B and C, the same three FAs determined the main differences between both groups. Considering the main FAs contributing most for the similarity among each of the defined groups, both groups A and B present palmitic acid, oleic acid, and EPA as the main FAs contributing for that similarity, in that order of contribution; group C presents exactly the same three FAs as the main contributors for the similarity of the group, but in a different order, with palmitic acid being, as well, the main contributor, but followed by EPA and, finally, oleic acid. The average similarities and dissimilarities among and between the groups, as well as the main three FAs contributing to the similarities and dissimilarities among and between the groups, are presented in Tables 5 and 6, respectively.

Considering each species individually, C16:0 was the most abundant fatty acid of *Laminaria ochroleuca*, contributing 21.09% of the species' total fatty acid content, followed by C18:1 (16.60%) and EPA and DHA in similar concentrations (13.84% and 13.58%, respectively). In *Saccharina latissima*, C16:0 was also the most abundant FA (contributing 19.88% of total FA), followed by C18:1 and EPA and C14:0 (18.70%, 12.87%, and 12.87 and 12.60%, respectively). *Sacchorizha polyschides* and *Undaria pinnatifida* showed C16:0 as the most abundant FA (contributing 19.10% of total FA content), followed by EPA (16.61%) and C18:1 (13.14%).

Table 7 indicates the total phenolic content of the kelps evaluated, where *S. polyschides* stands out for the highest content while *L. ochroleuca* revealed the lowest content. Overall, the content varied among the species studied, revealing statistically significant differences among each other.

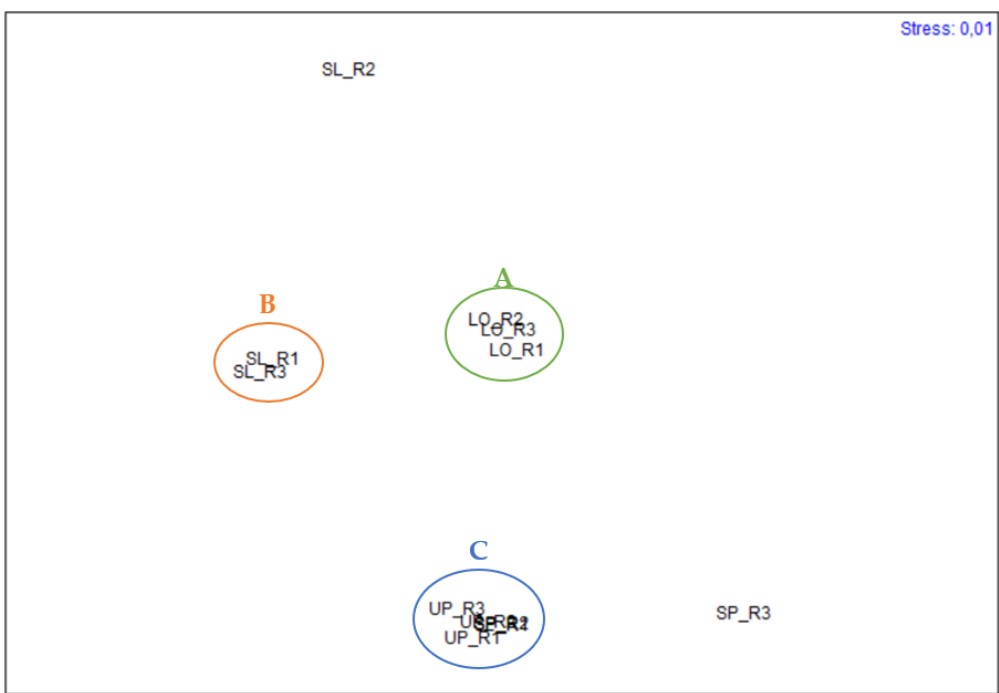

**Figure 2.** n-MDS of the four species studied, regarding the fatty acid profile: *Laminaria ochroleuca* (LO_R1, LO_R2, LO_R3), *Saccharina latissima* (SL_R1, SL_R2, SL_R3), *Sacchorriza polychides* (SP_R1, SP_R2, SP_R3), and *Undaria pinnatifida* (UP_R1, UP_R2, UP_R3). Group A is composed of LO_R1, LO_R2, and LO_R3 samples; group B is composed of SL_R1 and SL_R3 samples; Group C is composed of UP_R1, UP_R2, UP_R3, SP_R1, and SP_R2 samples.

**Table 5.** SIMPER results regarding similarities (sim.) within the groups identified (A—*Laminaria ochroleuca*; B—*Saccharina latissima*; C—*Sacchorizha polyschides* and *Undaria pinnatifida*). Similarities within groups are provided by the average similarity within each group (Av. Sim. within each group)–the higher the similarity (tending to 100), the higher the similarity within each group. The three FAs that contribute the most to the similarity in each group are shown, as well as the percentage of that contribution and the cumulative contribution of the three FAs for the total similarity of each group, in percentage. Replicates SL_R2 and SP_R3 were considered outliers of the analysis.

| Groups | Av. Sim. within Each Group | Main FA | % Contribution to Sim. | % Cumulative Contribution to Sim. |
|---|---|---|---|---|
| A | 95.92 | C16:0 | 31.71 | 65.00 |
|   |       | C18:1 | 19.64 |       |
|   |       | EPA   | 13.65 |       |
| B | 95.00 | C16:0 | 27.92 | 64.32 |
|   |       | C18:1 | 24.70 |       |
|   |       | EPA   | 11.70 |       |
| C | 92.76 | C16:0 | 26.97 | 60.22 |
|   |       | EPA   | 20.48 |       |
|   |       | C18:1 | 12.76 |       |

**Table 6.** SIMPER results regarding dissimilarities (diss.) between the groups identified (A—*Laminaria ochroleuca*; B—*Saccharina latissima*; C—*Sacchorizha polyschides* and *Undaria pinnatifida*). Dissimilarities between different groups are provided by the average dissimilarity within each group (Av. diss. between groups)—the lower the dissimilarity (tending to 0), the higher the similarity within each group. The three FAs that contribute the most for the dissimilarity between groups are shown, as well as the percentage of that contribution and the cumulative contribution of those three FAs for the total dissimilarity between groups, in percentage. Replicates SL_R2 and SP_R3 were considered outliers of the analysis.

| Groups | Av. Diss. between Groups | Main FA | % Contribution to Diss. | % Cumulative Contribution to Diss. |
|--------|------------------------|---------|------------------------|-----------------------------------|
| A, B | 14.17 | C18:1 | 26.80 | 62.60 |
|  |  | C14:0 | 23.09 |  |
|  |  | C16:1 | 12.70 |  |
| A, C | 20.57 | EPA | 32.89 | 67.00 |
|  |  | C18:3t | 19.07 |  |
|  |  | C18:3c | 15.04 |  |
| B, C | 26.17 | EPA | 26.61 | 59.58 |
|  |  | C18:3t | 19.37 |  |
|  |  | C18:3c | 13.60 |  |

**Table 7.** Total phenolic compound of the aqueous extracts of *U. pinnatifida*, *S. latissima*, *L. ochroleuca*, and *S. polyschides*. Results are expressed in mean $\pm$ standard deviation ($n = 3$; dry weight basis). Statistically significant differences among the species are expressed by different letters.

| Species | (g GAE/ 100 g Algae) |
|---------|----------------------|
| *U. pinnatifida* | $5.63 \times 10^{-3} \pm 1.43 \times 10^{-4}$ [a] |
| *S. polyschides* | $6.19 \times 10^{-3} \pm 1.15 \times 10^{-4}$ [b] |
| *S. latissima* | $4.91 \times 10^{-3} \pm 1.08 \times 10^{-4}$ [c] |
| *L. ochroleuca* | $3.72 \times 10^{-3} \pm 1.09 \times 10^{-4}$ [d] |

[a,b,c,d] Similar letters indicate no significant differences at the *p*-value < 0.05 level.

## 4. Discussion

The demand for functional foods is growing as people become more aware of the link between daily diet and health [41]. Functional foods and ingredients are useful not only because they contain vital nutrients but also because they contain bioactive substances, which have been discovered to be important for disease prevention and health improvement [42–45].

Seaweeds hold a rich nutritional profile as well as biologically active compounds, such as pigments, phenolic compounds, sulphated polysaccharides, fatty acids, and proteins, which could contribute to the development of several novel food products [41,42,46–49].

The habitat and the environment where seaweeds are present highly influences their biochemical and nutritional profile [50–52]. Nevertheless, seaweeds' chemical profile is species specific and varies according biotic and abiotic parameters variation [13,53,54]. Moreover, polysaccharides that constitute the seaweed cell wall have a substantial impact on its ability to absorb nutrients [55,56]. For this reason, brown seaweeds are well known for their exceptional mineral-absorption abilities [57,58]. Tables 8 and 9 sum the nutritional and macro and micro element analysis of the studied species reported in the literature.

The non-indigenous seaweed *U. pinnatifida* (or wakame) is among the most representative sea vegetable in the food market, holding a high economic value [59,60]. The biochemical profile of *U. pinnatifida*, harvested in Spain, was evaluated, exhibiting higher content of sodium and potassium (3511 and 5679 mg 100 g DW, respectively) but lower values of phosphorus, calcium, magnesium, copper, zinc, and manganese (1070, 693.2, 630.2, 0.19, 3.86, and 0.69 mg 100 g DW, respectively) [59]. However, the same seaweed harvested in Japan presented higher concentration of magnesium, copper, zinc, and manganese (405, 0.185, 0.944, and 0.332 mg 100 g DW, respectively) but lower content of



phosphorus, calcium, potassium, and sodium (450, 950, 5691, and 6494 mg 100 g DW, respectively) [61]. According to the literature, the nutritional profile of *U. pinnatifida* does not differ significantly across geographical zones, yielding similar results in terms of lipids, proteins, carbohydrates, fiber, and mineral content. The lipid and protein profiles, on the other hand, can vary or contain different concentrations of each compound [59,61], depending on the geographical distribution and respective environments.

The seaweed *S. polyschides* harvested from the central west coast of Portugal (Buarcos Bay, Figueira da Foz) in the spring of 2012 revealed an overall higher nutritional, micronutrient, and trace element composition [62]. As well as the seaweed *S. latissima*, which was harvested in the spring of 2010 in Nova Scotia, Canada, it exhibited an overall higher nutritional profile, presenting 8.1% of crude protein, 5.5% of total lipids, 24.5% of ashes, and 59.8% of carbohydrates [45]. Furthermore, the same species grown by a seaweed aquaculture company in northern France (Brittany) and harvested in April 2015 demonstrated a general higher concentration of the micronutrient and trace element composition. Despite the similar nutritional content of *S. latissima*, particularly regarding the lipid content, in this study, a lower protein concentration (10.2 g 100 g DW) and a higher ashes and fiber content (20.4 and 40.9 g 100 g DW, respectively) [32] was reported. However, it is important to remember that the harvesting season and geolocation, as well as the techniques used for these analyses, will all have an impact on the results obtained [63,64].

A team of researchers analyzed the chemical composition of *L. ochroleuca* harvested in the winter of 2015 in the Galician Vizcaya Gulf (Spain), revealing a lower micronutrient and trace element composition in comparison with the present study [65].

**Table 8.** Micronutrient and trace element composition of *U. pinnatifida*, *S. latissima*, *L. ochroleuca*, and *S. polyschides* according to the literature. ND—not determined.

| | % | | | | mg/100 g DW | | | | | | |
|---|---|---|---|---|---|---|---|---|---|---|---|
| | N | P | Ca | Mg | K | Na | Fe | Zn | Cu | Mn | Ref. |
| *U. pinnatifida* | ND | 450–1070 | 693–950 | 405–630 | 5679–5691 | 3511–6494 | ND | 0.94–3.86 | 0.944–3.86 | 0.33–0.69 | [59,61] |
| *S. polyschides* | 1–2.8 | 2.5 | 897–2207 | 240–1288 | 6576–18800 | 70 | 55.5–303 | 0.05 | 0.01 | 0.01 | [62,66] |
| *S. latissima* | 1.63 | ND | 919.4 | 611.1 | 3869.4 | 3048.3 | 185.4 | 3.86 | 3.86 | 0.56 | [32,45] |
| *L. ochroleuca* * | ND | 1.8 | 10.6 | 4.1 | 84.3 | 20.4 | 87.0 | 243.0 | 18.5 | ND | [65] |
| NVR (mg) | ND | 700 | 800 | 375 | 2000 | 600 | 14 | 10 | 1 | 2 | [67,68] |

* The results are expressed in mg/kg.

**Table 9.** Nutritional characterization of the dried biomass of *U. pinnatifida*, *S. latissima*, *L. ochroleuca*, and *S. polyschides* according to the literature. ND—not determined.

| | (g/100 g DW) | | | | | | |
|---|---|---|---|---|---|---|---|
| | Moisture | Ash | Lipid | Fiber | Carbohydrate | Protein | Reference |
| *U. pinnatifida* | ND | 26.58–37.58 | 1–3.13 | 2.7–14.9 | 35.3–50.4 | ND | [59,61,69] |
| *S. polyschides* | 10.88 | 28.15–58.8 | 0.3–0.9 | ND | 45.6 | 6.4–17.2 | [62,65] |
| *S. latissima* | ND | 24.3–27.3 | 0.10–5.5 | 6.2–7.1 | 60.3–66.8 | 7.4–11.7 | [32,45,70] |
| *L. ochroleuca* | ND | 9.47 | ND | ND | ND | 7.49 | [65,71] |
| NVR (g) | ND | ND | 70 | 25 | 260 | 50 | [67,68] |

In comparison with the nutrient reference value of a diet, established by the European Food Safety Authority (EFSA), a 100 g seaweed portion fulfills the needs of an average adult. However, the species *U. pinnatifida* and *S. polyschides* exceed the standard values (0.375 g), exhibiting an amount higher than 0.40 g Mg 100 g$^{-1}$. Moreover, all the seaweeds evaluated in the present study surpass the recommended amount of sodium (0.60 g), presenting values that exceed 1 g Na 100 g$^{-1}$. Regarding the iron concentration, the seaweeds *S. polyschides* and *S. latissima* also exceed the recommend values (0.014 g), containing 0.016 and 0.062 g Fe 100 g$^{-1}$. Nonetheless, seaweeds have the potential to be both functional foods and mineral nutraceutical suppliers [72]. The incorporation of these macro and micronutrients in the daily diet are pivotal because deficiencies in one or more of these nutrients can lead to chronic disorders in many circumstances [42]. For instance, calcium is a key structural bone and teeth component, and, for this reason, the amount of calcium

ingested as part of our diet has a significant impact on the prevention and mitigation of osteoporosis symptoms [73,74]. Iron is an important macronutrient since it is found in hemoglobin, responsible for the oxygen supply in the human body [75–77]. In addition, it is essential to ensure iron intake since it is a key component in DNA and RNA synthesis [12, 78]. This element is also involved in several metabolic processes, such as energy transport, and as a structural molecule in the formation of bones and membranes [79,80].

Potassium and sodium are vital electrolytes that aid the regulation of blood pressure, muscle contraction, heartbeat, and kidney function [81,82]. The imbalance of the intake of these nutrients often leads to the development of cardiovascular diseases [82,83]. Potassium is a particularly important element to stimulate kidney function [82,84,85].

The micronutrients zinc, copper, and manganese play critical roles in the human metabolism, as catalytic, structural, and regulatory processes, and are especially important for the neurological and cardiovascular systems [86–88].

Seasonal changes and geographical distribution may also contribute to the variation in the seaweed lipidic profile [89–91]. Although geographical differences may influence the genotypic expression and thus contribute to the potential phenotypic variability, they are primarily explained by phenotypic changes resulting from the environmental changes, mainly in water nutrients, light, and temperature, and may be meaningful even at minor spatial scales [64,92–94].

For instance, the seaweed *U. pinnatifida* harvested in central Portugal (Figueira da Foz, Buarcos bay) in January 2020 unveiled a different fatty acid profile, presenting the SFA C18:0, the MUFA C15:1, and the PUFA C20:4 [95], while *S. latissima* harvested in March of 2017 in Norway revealed a varied outline of fatty acids, presenting the SFA C17:0 and C18:0, as well as PUFA, such as C18:2 ω-6, C20:4 ω-6, and C18:4 ω-3 [96]. Other teams of researchers found that *Saccharina latissima* (July—summer), *Saccorhiza polyschides* (January—winter) and *Laminaria ochroleuca* (September—summer) collected in the north of Portugal (Viana do Castelo) presented a different fatty acid profile depending on the harvesting season [97]. In comparison with our results, these seaweeds revealed different qualitative and quantitative lipidic profiles [97].

Fatty acids are essential components of cell membranes that play crucial roles as energy sources in body growth and development [95,98,99]. Polyunsaturated fatty acids (PUFAs) ω-3 and ω-6 make up a significant amount of seaweed lipids, despite their modest crude lipid content. PUFAs are important elements of all cell membranes and precursors of eicosanoids, which are fundamental myriad regulators of different biological activities [100]. Moreover, PUFAs are considered health promoter agents due to their antimicrobial, anti-inflammatory, immunomodulatory, anticancer, and cardioprotective bioactivities [101,102], thus helping in the mitigation and prevention of diseases, such as obesity, Alzheimer's, osteoporosis, diabetes, and cardiovascular diseases [100,103–105]. More recently, it was found that the ω-6/ω-3 FA ratio in human diets is linked with the development of several autoimmune and cardiovascular diseases and diabetes [95,105,106]. In this context, a ratio of one is recognized as ideal for human diets [107]. Thus, the seaweeds in the current study, namely *L. ochroleuca*, *S. latissima*, *S. polyschides*, and *U. pinnatifida* present a suitable ω-6/ω-3 FA ratio (0.30, 0.19, 0.39, and 0.37, respectively) for their inclusion in the human diet.

Brown seaweeds' carbohydrates (or polysaccharides) are present in the algal cell wall, with the purpose of water and ion retention, in order to help them to cope with desiccation and osmotic stress during the low tide [108,109]. For this reason, the carbohydrate yield can highly differ according to the seaweed species and the variations of the biotic and abiotic factors [9,110,111].

In the food sector and for nutraceutical companies, algal polysaccharides are one of the most exploited constituents [5]. These compounds are used as natural food additives to improve food quality and as ingredients, principally as a dietary fiber [5,21]. Since seaweed polysaccharides behave as fiber when ingested by the human internal tract, they act as a probiotic food, preventing diseases, such as obesity, colorectal cancer, and gastrointestinal inflammation [42,112–114].

Proteins produced by seaweeds can vary between 5 and 47% on a dry basis, mostly due to the environmental parameter variations but also accordingly with the species [115]. The lack of proteins is a challenge to search for a new, low-cost protein source. Because of the relatively large quantity of nitrogen compounds in seaweeds, it could play a significant role in the aforementioned issue.

Thus, seaweeds can be used as a source of high-nutrition ingredients in the food industry [46]. However, to observe if the seaweeds targeted have good value as a raw food source, there is a need to observe the threshold values of the dried seaweed intake, which is normally 7 g daily, due to the iodine and other mineral content in seaweeds [7,9,10]. Moreover, there is a need to compare this content with terrestrial vegetables and to observe whether the food potential suppresses the need of vegetable consumption by humans.

Furthermore, using the known limiting factors, we calculated the maximum daily seaweed intake (LDI).

At first sight, the limiting factor in *U. pinnatifida* (Table 10), *S. polyschides* (Table 11), and *L. ochroleuca* (Table 12) is the sodium content, whereas in *S. latissima* (Table 13), it is the iron content. Thus, the mineral content is the major nutrient that demonstrates whether seaweed can be a sustainable source. *S. polyschides* is an important supplier of sodium, magnesium, iron, potassium, and calcium, while the other analyzed species have lower mineral content. The same mineral profile with lower quantities is present in *U. pinnatifida* and *L. ochroleuca*. Only the *S. latissima* has a different profile, with only iron, sodium, and magnesium showing interesting quantities but with low content in potassium and calcium when compared to the other seaweeds. For example, all analyzed species had higher amounts of iron (98 g) than broccoli or potatoes (4% of iron DRI—Dietary Reference Intake). The same was observed regarding sodium and potassium [116,117].

In terms of macronutrients, the main components of the analyzed seaweeds are fiber and protein, whereas 7 g of dried seaweed can fulfill between 1 to 3 percent of the Dietary Reference Intake (DRI). The brown seaweed *S. latissima* has a lower nutritional profile, while *L. ochroleuca* presents the highest percentage of fiber and *U. pinnatifida* the highest protein yield. In this context, the fiber content found in the analyzed seaweeds are similar to 39.47 g of broccoli (0.8 g of Fiber) or 26 g of sweet potato (0.8 g of Fiber). The protein profile is equivalent to 51.8 g of broccoli (1.4 g of protein) or 69.3 g of cauliflower (1.4 g of protein), indicating a viable terrestrial vegetable alternative [117].

Regarding calories intake, all the seaweeds show a low energy profile at 7 g or a limit daily intake (LDI) (Tables 9–12), where the maximum was registered in *L. ochroleuca* (7 g: 19.81 Kcal; LDI: 158.48 Kcal), demonstrating that seaweeds are a low caloric food source. For example, these species contain similar calories to 93 g of asparagus (20 Kcal), 148 g of tomato (25 Kcal), and lower than 67 g of lime or 147 g of peach. This demonstrates the possible impact of seaweeds in food security and as a key player in a nutritious food diet where vegetables and fruits do not satisfy all the needs of the human diet [5,117]. Thus, seaweeds can be key for the future sustainability of the food intake by humans, based on their potential in various macro- and micro-nutrients (as a food supplement) [5]. But, most important of all is that seaweeds do not compete by arable land and freshwater [5].

Nevertheless, the main risk to human life is the over intake of seaweed due to the low daily recommend dose of several micro- and macro-nutrients [118].

It is obvious that the seaweeds have a promissory potential to substitute vegetables, making a link between food security and supply, as possible food for the growing human population. There are also more benefits of using seaweeds as food supplements, such as health benefits of their components, turning them in nutraceuticals [5].

Furthermore, there are also secondary metabolites, which can enhance a positive impact on the human welfare. Moreover, the phenolic compounds are being studied, aiming for the mitigation or curing some of the most common medical concerns today, such as cardiovascular, diabetic, neurodegenerative, and mental disorders [25,42,119]. Algal phenolic compounds, especially phlorotannins, have exceptional antioxidant properties

due to their capacity to chelate reactive oxygen species, avoiding oxidative stress and cell damage as a result [119–121].

**Table 10.** Nutritional value in 7 g of *U. pinnatifida* according to the established Dietary Reference Intake (DRI) and its food intake limit using the DRI [5,9]. ND—not determined.

| Typology | Compound | 7 g | | Limit Daily Intake (42 g DW) | | |
| | | Quantity | % DRI | Quantity | % DRI | DRI |
|---|---|---|---|---|---|---|
| Micronutrient and trace element | Phosphorus (mg) | 15.5722 | 2.225% | 93.4332 | 13.348% | 700 |
| | Calcium (mg) | 42.6496 | 5.331% | 255.8976 | 31.987% | 800 |
| | Magnesium (mg) | 29.8655 | 7.964% | 179.193 | 47.785% | 375 |
| | Potassium (mg) | 38.7121 | 1.936% | 232.2726 | 11.614% | 2000 |
| | Sodium (mg) | 98.3178 | 16.386% | 589.9068 | 98.318% | 600 |
| | Iron (mg) | 0.4774 | 3.410% | 2.8644 | 20.460% | 14 |
| | Copper (mg) | 0.014 | 1.400% | 0.084 | 8.400% | 1 |
| | Zinc (mg) | 0.1939 | 1.939% | 1.1634 | 11.634% | 10 |
| | Manganese (mg) | 0.0266 | 1.330% | 0.1596 | 7.980% | 2 |
| Nutrients | Ash (g) | 1.3006 | ND | 7.8036 | ND | ND |
| | Total lipid (g) | 0.0441 | 0.063% | 0.2646 | 0.378% | 70 |
| | Fatty Acids (mg) | 245.91 | 0.56% | 1475.46 | 3.35% | 44,000 |
| | Fiber (g) | 0.616 | 2.464% | 3.696 | 14.784% | 25 |
| | Protein (g) | 1.6478 | 3.296% | 9.8868 | 19.774% | 50 |
| | Nitrogen-Free Extractives (g) | 3.3908 | 1.304% | 20.3448 | 7.825% | 260 |
| | Energy (Kcal) | 20.58 | 1.029% | 123.48 | 6.174% | 2000 |

**Table 11.** Nutritional value in 7 g of *S. polyschides* according to the established Dietary Reference Intake (DRI) and its food intake limit using the DRI [5,9]. ND—not determined.

| Typology | Compound | 7 g | | Limit Daily Intake (54 g DW) | | |
| | | Quantity | % DRI | Quantity | % DRI | DRI |
|---|---|---|---|---|---|---|
| Micronutrient and trace element | Phosphorus (mg) | 8.1935 | 1.171% | 63.207 | 9.030% | 700 |
| | Calcium (mg) | 48.2503 | 6.031% | 372.2166 | 46.527% | 800 |
| | Magnesium (mg) | 31.3075 | 8.349% | 241.515 | 64.404% | 375 |
| | Potassium (mg) | 108.0772 | 5.404% | 833.7384 | 41.687% | 2000 |
| | Sodium (mg) | 76.5947 | 12.766% | 590.8734 | 98.479% | 600 |
| | Iron (mg) | 1.1235 | 8.025% | 8.667 | 61.907% | 14 |
| | Copper (mg) | 0.0161 | 1.610% | 0.1242 | 12.420% | 1 |
| | Zinc (mg) | 0.1834 | 1.834% | 1.4148 | 14.148% | 10 |
| | Manganese (mg) | 0.0322 | 1.610% | 0.2484 | 12.420% | 2 |
| Nutrients | Ash (g) | 1,4623 | ND | 11.2806 | ND | ND |
| | Total lipid (g) | 0,105 | 0.150% | 0.81 | 1.157% | 70 |
| | Fatty Acids (mg) | 229.6 | 0.52% | 1771.2 | 4.03% | 44,000 |
| | Fiber (g) | 0.7378 | 2.951% | 5.6916 | 22.766% | 25 |
| | Protein (g) | 0.9807 | 1.961% | 7.5654 | 15.131% | 50 |
| | Nitrogen-Free Extractives (g) | 3.7135 | 1.428% | 28.647 | 11.018% | 260 |
| | Energy (Kcal) | 19.74 | 0.987% | 152.28 | 7.614% | 2000 |

**Table 12.** Nutritional value in 7 g of *L. ochroleuca* according to the established Dietary Reference Intake (DRI) and its food intake limit using the DRI [5,9]. ND—not determined.

| Typology | Compound | 7 g | | Limit Daily Intake (56 g DW) | | DRI |
|---|---|---|---|---|---|---|
| | | Quantity | % DRI | Quantity | % DRI | DRI |
| Micronutrient and trace element | Phosphorus (mg) | 9.233 | 1.319% | 73.864 | 10.552% | 700 |
| | Calcium (mg) | 46.0649 | 5.758% | 368.5192 | 46.065% | 800 |
| | Magnesium (mg) | 18.6263 | 4.967% | 149.0104 | 39.736% | 375 |
| | Potassium (mg) | 86.5487 | 4.327% | 692.3896 | 34.619% | 2000 |
| | Sodium (mg) | 74.9182 | 12.486% | 599.3456 | 99.891% | 600 |
| | Iron (mg) | 0.581 | 4.150% | 4.648 | 33.200% | 14 |
| | Copper (mg) | 0.0217 | 2.170% | 0.1736 | 17.360% | 1 |
| | Zinc (mg) | 0.133 | 1.330% | 1.064 | 10.640% | 10 |
| | Manganese (mg) | 0.0882 | 4.410% | 0.7056 | 35.280% | 2 |
| Nutrients | Ash (g) | 1.2831 | ND | 10.2648 | ND | ND |
| | Total lipid (g) | 0.0357 | 0.051% | 0.2856 | 0.408% | 70 |
| | Fatty Acids (mg) | 167.09 | 0.38% | 1336.72 | 3.04% | 44,000 |
| | Fiber (g) | 0.8155 | 3.262% | 6.524 | 26.096% | 25 |
| | Protein (g) | 0.8981 | 1.796% | 7.1848 | 14.370% | 50 |
| | Nitrogen-Free Extractives (g) | 3.9676 | 1.526% | 31.7408 | 12.208% | 260 |
| | Energy (Kcal) | 19.81 | 0.991% | 158.48 | 7.924% | 2000 |

**Table 13.** Nutritional value in 7 g of *S. latissima* according to the established Dietary Reference Intake (DRI) and its food intake limit using the DRI [5,9]. ND—not determined.

| Typology | Compound | 7 g | | Limit Daily Intake (22 g DW) | | DRI |
|---|---|---|---|---|---|---|
| | | Quantity | % DRI | Quantity | % DRI | DRI |
| Micronutrient and trace element | Phosphorus (mg) | 7.7336 | 1.105% | 24.3056 | 3.472% | 700 |
| | Calcium (mg) | 31.0989 | 3.887% | 97.7394 | 12.217% | 800 |
| | Magnesium (mg) | 19.8002 | 5.280% | 62.2292 | 16.594% | 375 |
| | Potassium (mg) | 175.938 | 8.797% | 552.948 | 27.647% | 2000 |
| | Sodium (mg) | 88.004 | 14.667% | 276.584 | 46.097% | 600 |
| | Iron (mg) | 4.3855 | 31.325% | 13.783 | 98.450% | 14 |
| | Copper (mg) | 0.0371 | 3.710% | 0.1166 | 11.660% | 1 |
| | Zinc (mg) | 0.1981 | 1.981% | 0.6226 | 6.226% | 10 |
| | Manganese (mg) | 0.0315 | 1.575% | 0.099 | 4.950% | 2 |
| Nutrients | Ash (g) | 1.2467 | ND | 3.9182 | ND | ND |
| | Total lipid (g) | 0.1064 | 0.152% | 0.3344 | 0.478% | 70 |
| | Fatty Acids (mg) | 158.13 | 0.36% | 496.98 | 1.13% | 44,000 |
| | Fiber (g) | 0.4473 | 1.789% | 3.9182 | 5.623% | 25 |
| | Protein (g) | 0.9541 | 1.908% | 0.3344 | 5.997% | 50 |
| | Nitrogen-Free Extractives (g) | 4.2448 | 1.633% | 3.9182 | 5.131% | 260 |
| | Energy (Kcal) | 21.77 | 1.089% | 68.42 | 3.421% | 2000 |

## 5. Conclusions

Seaweeds are seen as promising organisms for delivering essential compounds for human nutrition in recent decades, with potentially significant economic influences on the food industry and on human health.

An adequate daily intake of minerals is required for the normal functioning of the human organism; from this perspective, seaweeds are an important feedstock for the food sector.

Phytochemicals that show antioxidant, antibacterial, anticancer, and antiviral action found in seaweeds are responsible for these therapeutic effects. Phenolic compounds,

sulphated polysaccharides, and organic acids, to name a few, also participate in these actions. In this context, seaweeds, or their extracts, offer functional properties that are increasingly being used in diets and take advantage of these benefits.

However, much more research is needed, such as on seaweed's role in nutrition and disease prevention, before science-based dietary recommendations for edible seaweeds can be made. In this regard, this study concludes that the brown seaweeds *Saccorhiza polyschides*, *Laminaria ochroleuca*, *Saccharina latissima*, and *Undaria pinnatifida* contain a rich nutritional value, with nutraceutical potential.

**Author Contributions:** Conceptualization, D.P., J.C., A.M.M.G., K.B. and L.P.; methodology, D.P., G.M., C.P.R., R.L.P., J.C., A.M.M.G., S.M.D.S., K.B. and L.P.; nutritional characterization, D.P., G.M., C.P.R., R.L.P. and J.C.; fatty acid analysis: C.P.R. and A.M.M.G.; investigation, D.P., G.M., C.P.R., R.L.P. and J.C.; data curation, D.P., G.M., C.P.R., R.L.P. and J.C.; writing—original draft preparation, D.P., G.M., J.C. and C.P.R.; writing—review and editing, D.P., G.M., C.P.R., R.L.P., J.C., A.M.M.G., S.M.D.S., K.B. and L.P.; supervision, A.M.M.G., S.M.D.S., K.B. and L.P. All authors have read and agreed to the published version of the manuscript.

**Funding:** This work was financed by national funds through the FCT (Foundation for Science and Tech-nology), I.P., within the scope of the projects UIDB/ 04292/2020 (MARE, Marine and Environmental Sciences Centre) and UIDP/50017/2020+UIDB/50017/2020 (CESAM, Centre for Environmental and Marine Studies).

**Institutional Review Board Statement:** Not applicable.

**Informed Consent Statement:** Not applicable.

**Data Availability Statement:** Not applicable.

**Acknowledgments:** The project PORBIOTA—E-Infrastructure Portuguese Information and Research in Biodiversity (POCI-01-0145-FEDER-022127) co-financed this research, supported by the Competitiveness and Internationalization Operational Programme and Regional Operational Programme of Lisbon, through FEDER, and by the Portuguese Foundation for Science and Technology (FCT), through national funds (OE). Diana Pacheco offers thanks to PTDC/BIA-CBI/ 31144/2017-POCI-01 project-0145-FEDER-031144-MARINE INVADERS, co-financed by the ERDF through POCI (Operational Program Competitiveness and Internationalization) and by the Foundation for Science and Technology (FCT, IP). João Cotas offers thanks to the European Regional Development Fund through the Interreg Atlantic Area Pro-gram, under the project NASPA (EAPA_451/2016). Ana M. M. Gonçalves acknowledges the University of Coimbra for the contract IT057-18-7253 and also offers thanks to the project MENU—Marine Macroalgae: Alternative recipes for a daily nutritional diet (FA_05_2017_011), funded by the Blue Fund under Public Notice No. 5—Blue Biotechnology.

**Conflicts of Interest:** The authors declare no conflict of interest.

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
