# Peer review of "Portuguese Kelps: Feedstock Assessment for the Food Industry"

_applsci, doi:10.3390/app112210681_

Round 1

Reviewer 1 Report

The manuscript is  concerned analyses of nutritional components of four brown algae.

In proximate determination, the authors represent average and standard deviation values of only twice measurements (n=2). This interpretation is unreasonable. The authors should increase number of determination.

The authors determined fatty acid composition of the algal lipids. To the best of my knowledge, brown algae exhibit little amount of EPA and DHA contents, and considerable amount of C18:4. The authors should re-examine GC-MS analysis. 

In discussion section, the content is redundant. This manuscript is submitted as an article, not a review. The authors should edit the section.

Author Response

Reviewer 1:

Comment 1: The manuscript is concerned analyses of nutritional components of four brown algae.

In proximate determination, the authors represent average and standard deviation values of only twice measurements (n=2). This interpretation is unreasonable. The authors should increase number of determination.

Answer 2: Due to the biomass available and logistics, only two analyses were possible to perform. Still, from the authors’ experience the data present are in accordance to the information in literature and expected.

Comment 3: The authors determined fatty acid composition of the algal lipids. To the best of my knowledge, brown algae exhibit little amount of EPA and DHA contents, and considerable amount of C18:4. The authors should re-examine GC-MS analysis.

Answer 3: The authors thank the comment and have, indeed, re-examined the GC-MS analysis, having revised some values, however, the pattern found (of considerable EPA and DHA amounts and no C18:4 content) was maintained, as firstly presented. The authors discuss that this fact may be explained by the variability of algae fatty acid profile depending on the set of environmental factors the organisms are subjected to, varying between zones and seasons, for example, as reported elsewhere (eg. Susanto et al., 2016. Doi: https://doi.org/10.1016/j.aqpro.2016.07.009).

Comment 4: In discussion section, the content is redundant. This manuscript is submitted as an article, not a review. The authors should edit the section.

Answer 4: The purpose of this study is not only to assess the nutritional value of Portuguese kelps, but also to determine how they might contribute to the food market through their nutraceutical potential. As a result, how can seaweed minerals and fatty acids contribute to the Dietary Reference Intake defined by competent authorities and how can they contribute to human health is discussed. By this, the author would like to maintain the section as it is.

Reviewer 2 Report

The manuscript entitled “Portuguese Kelps: feedstock assessment for the food industry” is suitable for the publication in Foods after the major revision.

The results are interesting, but there are few things to improve before acceptation

  • the aim of the work should be more specific - it is mentioned that diet affects the occurrence of diseases such as diabetes and obesity. In this case, it should be more precisely indicated which components of kelp can prevent this and what has been studied in the work
  • in addition, it should be highlighted what is new in this work - since, for example, the exact composition of these species is given in the tables (tab. 9-12) in the discussion
  • the results in table 1 - please check carefully - the authors write that the results are mean ± standard deviation - is it really possible? in the case of many elements the deviations are much higher than the average - how is this possible?
  • I think that the composition of fresh plants should be given first and then dried ones, i.e. the sequence of tables 2 and 3 should be the opposite
  • please check the statistical analysis of fig. 2 carefully. Standard deviations are very large, therefore it is doubtful whether in the case of C18: 3 the difference between SP and UP is statistically significant, as well as DHA. Apart from tm, the abbreviations used in Figure 2 and Table 4 are not explained
  • Fig 3, despite the use of colors, is difficult to read
  • Please explain in more detail how the similarities and differences of groups A, B and C were shown (tab. 5a and 5b).
  • the content of polyphenols alone does not contribute much, it would be worth analyzing their composition, or at least defining the subclass (flavonoids, phenolic acids?)
  • in the reference list, entries 122 and 125 are a repetition

Author Response

Reviewer 2:

Comment 1:The manuscript entitled “Portuguese Kelps: feedstock assessment for the food industry” is suitable for the publication in Foods after the major revision.

The results are interesting, but there are few things to improve before acceptation

Answer 1: The authors appreciate the reviewer's kind words.

Comment 2: The aim of the work should be more specific - it is mentioned that diet affects the occurrence of diseases such as diabetes and obesity. In this case, it should be more precisely indicated which components of kelp can prevent this and what has been studied in the work

Answer 2: According to the suggestions, the authors have rewritten the goal of the study.

Comment 3:in addition, it should be highlighted what is new in this work - since, for example, the exact composition of these species is given in the tables (tab. 9-12) in the discussion

Answer 3: The composition of algal biomass can vary greatly depending on biotic and abiotic conditions. As a result, it is critical to describe algal biomass in various places, which is critical for food sector supply. Tables 9-12 were created using the mineral content values and the daily limit imposed by the competent authorities, in order to understand how kelps biomass can contribute to the input of each nutrient.

Comment 4: the results in table 1 - please check carefully - the authors write that the results are mean ± standard deviation - is it really possible? in the case of many elements the deviations are much higher than the average - how is this possible?

Answer 4: The authors double-checked the results and discovered that the standard deviations were incorrect due to the conversions. Thank you for bringing this to our attention. The standard deviations had been corrected by the authors.

Comment 5: I think that the composition of fresh plants should be given first and then dried ones, i.e. the sequence of tables 2 and 3 should be the opposite

Answer 5: Thank you for your suggestion, the authors inverted the order of the tables and the text.

Comment 6: please check the statistical analysis of fig. 2 carefully. Standard deviations are very large, therefore it is doubtful whether in the case of C18: 3 the difference between SP and UP is statistically significant, as well as DHA. Apart from tm, the abbreviations used in Figure 2 and Table 4 are not explained

Answer 6: The authors thank the comment and the correction. There was, indeed, a mistake in the letters indicating statistical differences for C18:1 and DHA – no statistically significant differences were found among species for these FA. The letter “b” indicated in the bars corresponding to the two FA was left unintendedly. The abbreviations in Figure 2 and Table 4 have been included in the respective captions.

Comment 7: Fig 3, despite the use of colors, is difficult to read

Answer 7: The authors acknowledge the reviewer’s comment, but, indeed, after several attempts to improve the quality of the image and the relation between the labels corresponding to the samples, the provided figure was the best we could export from the software. To overcome the difficulty in reading the labels, we have opted to indicate, in the figure’s caption, the composition of each group.

Comment 8: Please explain in more detail how the similarities and differences of groups A, B and C were shown (tab. 5a and 5b).

Answer 8: A further explanation of the interpretation of tables 5a and 5b has been added to each table caption.

Comment 9: The content of polyphenols alone does not contribute much, it would be worth analyzing their composition, or at least defining the subclass (flavonoids, phenolic acids?)

Answer 9: It was not possible to assess their composition due to logistical constraints. The concentration was too low to be analyzed spectrophotometrically.

Comment 10: in the reference list, entries 122 and 125 are a repetition

Answer 10: Thank you for bringing this to our attention. It has been addressed.

Reviewer 3 Report

Dear Authors,

The search for alternative nutritional sources is one of the most relevant research areas. Seaweeds are attractive source of nutrients and bioactive substances with various biological properties.  Authors assessed the nutritional perspectives of brown seaweeds Undaria pinnatifida, Saccharina latissima, Sacchoriza polyschides and Laminaria ochroleuca. Although the topic in this work was interesting, the presentation in this manuscript was insufficiently clear. Here are my detailed comments:

  1. In Abstract it worth be mentioned research methods in brief.
  2. Page 1, lines 24-24: “Significant concentrations of fatty acids were observed in both seaweeds, with U. pinnatifida having the highest value (10.20 mg/g DW) and S. latissima the lowest 25 content (4.81 mg/g DW)”. Do you mean HUFA instead fatty acids?
  3. Page 3, line 91: “…..food-related diseases such as diabetes and obesity”. What compounds in algae spp. could influence on this diseases?
  4. Section 2. Materials and Methods. Please, wherever weighing was used, specify the scales (Manufacturer, the country of manufacturer).
  5. Section 2. Materials and Methods. Please, wherever calculations were made, used equation should be given, as well as units in which the results were expressed.
  6. Section 2. Materials and Methods. The manufacturer and the country of manufacture for reagents and equipment are not indicated everywhere:

- Page 4, line 135: methanol

- Page 4, line 140: column

- Page 4, line 146: detector

- Page 4, line 150: software

- Page 4, line 163: Kjeldhal distiller

- Page 4, line 167: sodium hydroxide

- Page 4, line 180: fiber analyzer

- Page 5, line 194: nitric acid

- Page 5, line 203: Buchner and Goosh filter

- Page 5, line 207: Folin-Ciocalteu reagent

- Page 5, line 208: sodium carbonate

  1. In Subsection 2.3.1. Page 4, line 137. Please, indicate the condition of centrifugation (g, time, temperature) and equipment.
  2. In Subsection 2.7. Why do you use distilled water for extraction? Organic solvents are usually used, such as methanol or ethanol. Do you use a standard curve? Please, mention the concentration range for standard curve.
  3. In Subsection 2.8. How the data was expressed? Mean±SE or SD?
  4. Please, carefully check the results in the text (Page 6, lines 235-245) with the table 1. For e.g. Page 6, line 235 “and sodium (3.8 % DW, 0.223 g P 100g-1 and 1.404 g K 100g-1, respectively)”, while in table 1.40454 and Na instead 1.404 g K 100g-1.
  5. Page 6, line 235. Does the nitrogen is expressed in %DW, if so, it should be mentioned in Table 1.
  6. Please, describe below the Tables 1-4,6 and Figure 2 the meaning of letters “a-c”. It will be worth for understanding, if for e.g. letters in pair “a-b” mean difference between certain pair.
  7. Page 6, line 251: “Dry (Table 2) and fresh (Table 3)…” In Section 2. Materials and Methods. there was no mention about analyzing fresh matter.
  8. Concerning describing results. Please, be careful with describing the results close in value. For e.g. Page 6, lines 256-257: “S. polyschides stands out for the lowest energy value, where a 100 g portion provides 1180 KJ”. As it is in Table 2, the energy value of L. ochroleuca is 1183 KJ, which is approximately similar with S. polyschides. A similar description is found throughout the text in section 3. Results. As advise, it worth be describing only statistically different values.
  9. Table 3. describes the fresh biomass, but the results expressed in 100g-1 DW. Please, check it.
  10. Page 7, line 287: “…per gram of algae” DW or FW?
  11. Table 4. The total FA content is 23.87, 22.59, 32.80 and 35.13 mg/g, while in Table 2 the fat content 0.51, 1.52, 1.50 and 0.63 g/100gDW, which correspond 5.1, 15.2, 15.0 and 6.3 mg/g DW. Please check the results or explain the FA content exceeding compared with fat content.
  12. Table 4. Please, insert the statistical differences.
  13. Table 6. The other results expressed on g or 100 g, why Total phenolic compound is expressed on L? It worth be calculated on g or 100 g.
  14. Page 12, line 395: “0.19”, but in Table 7 – ND.
  15. Page 12, line 408: “5.5%”, but in Table 8 – 0.10-0.12 g/100g.
  16. Page 12, lines 413-414: “0.19”. It is not clear for which species data are given.
  17. Page 14, line 513. Mentioning of Table 12 is missed.
  18. Tables 9-12. To improve understanding of tables, it is recommended adding the DRI values.

Best regards,

Reviewer.

Author Response

Reviewer 3:

Comment 1: The search for alternative nutritional sources is one of the most relevant research areas. Seaweeds are attractive source of nutrients and bioactive substances with various biological properties.  Authors assessed the nutritional perspectives of brown seaweeds Undaria pinnatifida, Saccharina latissima, Sacchoriza polyschides and Laminaria ochroleuca. Although the topic in this work was interesting, the presentation in this manuscript was insufficiently clear. Answer 1: The authors appreciate the reviewer's comments on how and where to improve the manuscript.

Comment 2: In Abstract it worth be mentioned research methods in brief.

Answer 2: According to the suggestion, it was added in the abstract a brief description of the methods employed on the seaweed’s biomass analysis.

Comment 3: Page 1, lines 24-24: “Significant concentrations of fatty acids were observed in both seaweeds, with U. pinnatifida having the highest value (10.20 mg/g DW) and S. latissima the lowest 25 content (4.81 mg/g DW)”. Do you mean HUFA instead fatty acids?

Answer 3: Thank you for bringing this to our attention. It has been corrected.

Comment 4: Page 3, line 91: “…..food-related diseases such as diabetes and obesity”. What compounds in algae spp. could influence on this diseases?

Answer 4: According to the authors quoted, the total phenolic content of these brown seaweeds can decrease glucose absorption. Furthermore, seaweeds contain carbohydrates that the human digestive tract is unable to degrade, aiding in the fight against obesity. This is also highlighted in the discussion.

Comment 5: Section 2. Materials and Methods. Please, wherever weighing was used, specify the scales (Manufacturer, the country of manufacturer).

Answer 5: Thank you for bringing this to our attention. It has been addressed.

Comment 6: Section 2. Materials and Methods. Please, wherever calculations were made, used equation should be given, as well as units in which the results were expressed.

Answer 6: It was provided the formulas used to calculate each parameter.

Comment 7: Section 2. Materials and Methods. The manufacturer and the country of manufacture for reagents and equipment are not indicated everywhere:

 Page 4, line 135: methanol

- Page 4, line 140: column

- Page 4, line 146: detector

- Page 4, line 150: software

- Page 4, line 163: Kjeldhal distiller

- Page 4, line 167: sodium hydroxide

- Page 4, line 180: fiber analyzer

- Page 5, line 194: nitric acid

- Page 5, line 203: Buchner and Goosh filter

- Page 5, line 207: Folin-Ciocalteu reagent

- Page 5, line 208: sodium carbonate

Answer 7: Thank you for alerting us to this. It was resolved properly.

Comment 8: In Subsection 2.3.1. Page 4, line 137. Please, indicate the condition of centrifugation (g, time, temperature) and equipment.

Answer 8: It was added in Subsection 2.3.1. the information missing.

Comment 9: In Subsection 2.7. Why do you use distilled water for extraction? Organic solvents are usually used, such as methanol or ethanol. Do you use a standard curve? Please, mention the concentration range for standard curve.

Answer 9: The authors are aware that the extraction of total phenolic compounds with other solvents, such as ethanol or methanol, may be higher. However, due to its use in the food industry, aqueous extraction reveals to be the safest for further food application. Yes, a standard curve was used. The concentrations used to achieve this goal were added in subsection 2.7.

Comment 10: In Subsection 2.8. How the data was expressed? Mean±SE or SD?

Answer 10: The results are expressed by mean ± standard deviation, and this information was added in the subsection 2.8.

Comment 11: Please, carefully check the results in the text (Page 6, lines 235-245) with the table 1. For e.g. Page 6, line 235 “and sodium (3.8 % DW, 0.223 g P 100g-1 and 1.404 g K 100g-1, respectively)”, while in table 1.40454 and Na instead 1.404 g K 100g-1.

Answer 11: The authors double-checked the information in the tables and the text, making the necessary changes.

Comment 12: Page 6, line 235. Does the nitrogen is expressed in %DW, if so, it should be mentioned in Table 1.

Answer 12: This was corrected in text; the nitrogen is expressed in g Na 100g-1.

Comment 13: Please, describe below the Tables 1-4,6 and Figure 2 the meaning of letters “a-c”. It will be worth for understanding, if for e.g. letters in pair “a-b” mean difference between certain pair.

Answer 13: A table footer and image footer were added with the meaning of the letters.

Comment 14: Page 6, line 251: “Dry (Table 2) and fresh (Table 3)…” In Section 2. Materials and Methods. there was no mention about analyzing fresh matter.

Answer 14: The authors had already corrected the table caption as it was a typo.

Comment 15: Concerning describing results. Please, be careful with describing the results close in value. For e.g. Page 6, lines 256-257: “S. polyschides stands out for the lowest energy value, where a 100 g portion provides 1180 KJ”. As it is in Table 2, the energy value of L. ochroleuca is 1183 KJ, which is approximately similar with S. polyschides. A similar description is found throughout the text in section 3. Results. As advise, it worth be describing only statistically different values.

Answer 15: The authors had revised the text and made the necessary changes.

Comment 16: Table 3. describes the fresh biomass, but the results expressed in 100g-1 DW. Please, check it.

Answer 16: This was revised and corrected.

Comment 17: Page 7, line 287: “…per gram of algae” DW or FW?

Answer 17: In this case, it’s referring to the dried algae. This information was added in the text.

Round 2

Reviewer 1 Report

The revised manuscript is not responded to the previous reviewer's comment. For example, I cannot understand to represent standard deviation calculated from several data.

Author Response

Reviewer 1:

Comment 1: The revised manuscript is not responded to the previous reviewer's comment. For example, I cannot understand to represent standard deviation calculated from several data.

Answer 1: The authors addressed the previous comments from the reviewers. Nonetheless, in the subsection 2.8. was added with the information about results expression for all measurements. Moreover, the standard deviation was calculated for all measurements, to understand how spread out the data is.

Reviewer 2 Report

 In my opinion the authors have corrected the manuscript in line with the comments and it can be accepted.

Author Response

Reviewer 2:

Comment 1: In my opinion the authors have corrected the manuscript in line with the comments and it can be accepted.

Answer 1: The authors acknowledge the kind words from the reviewer and we are grateful for your insightful comments on the manuscript.

Reviewer 3 Report

Dear Authors,

Thank You for careful revising the manuscript.

Here are my additional comments:

  1. Page 1, line 20: “total fiber content fiber” – duplicate word. Please, check it.
  2. Subsection 2.3.1. Page 5, line 185. Please, indicate the condition of centrifugation (g, time, temperature of centrifugation) and equipment. The information is still missing.
  3. Section 2. Materials and Methods. The manufacturer and the country of manufacture for equipment are still not indicated for following items:

- Page 5, line 189: column

- Page 4, line 195: detector

- Page 4, line 199: software

  1. The information about results expression should be given for all measurements and noted in Subsection 2.8. “Statistical Analysis”, but not in Subsection 2.7. (Page 7, Lines 289-290). Please, provide Subsection 2.8. with this information.
  2. Page 8, line 313 “and sodium (………. g K 100g-1, respectively)”, Na instead K? Please, check it.
  3. As proposal, the values in the Table 1 should be the similar with the text description (in text the values is rounded).
  4. Page 9, lines 351-353: “Comparatively with the other analyzed seaweeds, S. latissima stands out positively for having the highest amount of lipids or fat (1.52 g 100g-1)”, but the value for S. polyschides is approximately similar (1.50). Please, check the results description.
  5. Unfortunately, there were no answers in the Web-system for Comments 18-25, but the changes in the manuscript have been made. However, I’d like to see the answers for following previous comments:

- The total FA content is 23.87, 22.59, 32.80 and 35.13 mg/g, while in Table 2 the fat content 0.51, 1.52, 1.50 and 0.63 g/100gDW, which correspond 5.1, 15.2, 15.0 and 6.3 mg/g DW. Please check the results or explain the FA content exceeding compared with fat content. The values for total FA content were changed but still higher than in Table 2. Please, explain it.

- Table 4. Please, insert the statistical differences.

- Table 6. The other results expressed on g or 100 g, why Total phenolic compound is expressed on L? It worth be calculated on g or 100 g.

-Lines 482-484: “but lower values of phosphorus, calcium, magnesium, copper, zinc and manganese (1070, 693.2, 0.19, 3.86 and 0.69 mg/ 100g DW, accordingly)”. There are 6 items and 5 values. Please, check it.

Best regards,

Reviewer.

Author Response

Reviewer 3:

Comment 1: Here are my additional comments:

  1. Page 1, line 20: “total fiber content fiber” – duplicate word. Please, check it.
  2. Subsection 2.3.1. Page 5, line 185. Please, indicate the condition of centrifugation (g, time, temperature of centrifugation) and equipment. The information is still missing.
  3. Section 2. Materials and Methods. The manufacturer and the country of manufacture for equipment are still not indicated for following items:

- Page 5, line 189: column

- Page 4, line 195: detector

- Page 4, line 199: software

Answer 1: Thank you for pointing out these issues, the additional information was incorporated in the material and methods section.

Comment 2: The information about results expression should be given for all measurements and noted in Subsection 2.8. “Statistical Analysis”, but not in Subsection 2.7. (Page 7, Lines 289-290). Please, provide Subsection 2.8. with this information.

  1. Page 8, line 313 “and sodium (………. g K 100g-1, respectively)”, Na instead K? Please, check it.
  2. As proposal, the values in the Table 1 should be the similar with the text description (in text the values is rounded).
  3. Page 9, lines 351-353: “Comparatively with the other analyzed seaweeds, S. latissima stands out positively for having the highest amount of lipids or fat (1.52 g 100g-1)”, but the value for S. polyschides is approximately similar (1.50). Please, check the results description.

Answer 2: In the subsection 2.8. was added with the information about results expression for all measurements. Regarding the corrections on lines 313, 351-353, the authors revised and corrected the manuscript. Regarding the text of Table 1, the reviewer suggestion was followed and incorporated in the manuscript.

Comment 3: Unfortunately, there were no answers in the Web-system for Comments 18-25, but the changes in the manuscript have been made. However, I’d like to see the answers for following previous comments:

- The total FA content is 23.87, 22.59, 32.80 and 35.13 mg/g, while in Table 2 the fat content 0.51, 1.52, 1.50 and 0.63 g/100gDW, which correspond 5.1, 15.2, 15.0 and 6.3 mg/g DW. Please check the results or explain the FA content exceeding compared with fat content. The values for total FA content were changed but still higher than in Table 2. Please, explain it.

- Table 4. Please, insert the statistical differences.

- Table 6. The other results expressed on g or 100 g, why Total phenolic compound is expressed on L? It worth be calculated on g or 100 g.

-Lines 482-484: “but lower values of phosphorus, calcium, magnesium, copper, zinc and manganese (1070, 693.2, 0.19, 3.86 and 0.69 mg/ 100g DW, accordingly)”. There are 6 items and 5 values. Please, check it.

Answer 3:

- The authors have rechecked the results and have revised some values, however, the relation between total fatty acids and fat content was maintained. The authors discussed this could be due to the different methods performed for the determination of total fat and total fatty acids, which were, as described, analysed independently. Indeed, the temperature used for the Soxhlet method could have degraded some FA and prevented its inclusion in the final fat content. Moreover, the method used for the extraction of fatty acid for GC-MS analysis is more refined, being a more sensitive technique.

- Regarding the table 4, the statistical differences are in the Figure 2, so the authors think that there is no need to repeat the information.

- Because the analysis could only be performed directly on the aqueous extracts, the results are expressed in mg GAE/ L.

- The information of the lines 482-484 were revised and corrected.

Round 3

Reviewer 1 Report

The authors must not use standard deviation calculated from 2 sample size. Please check difference between standard deviation and standard error of mean. The authors should change representations of results.

Author Response

Reviewer 1:

Comment 1: The authors must not use standard deviation calculated from 2 sample size. Please check difference between standard deviation and standard error of mean. The authors should change representations of results.

Answer 1: As suggested, the representation of the data of table 1-3 was modified, within the standard error of mean.

Reviewer 3 Report

Dear Authors,

Thank You for careful revising the manuscript.

Here are my additional comments:

  1. Subsection 2.3.1. Page 5, line 186. The “rpm” should be converted in “g”. Please, recalculate “rpm” in “g”.
  2. Do I understand correctly that the data in Table 4 and Figure 2 duplicate each other? If so, please, leave Table 4 or Figure 2, not both.
  3. - Table 6. The other results expressed on g or 100 g, why Total phenolic compound is expressed on L? The analysis could only be performed directly on the aqueous extracts, but a certain sample (g or mg) of algae was used for preparing of aqueous extracts, therefore the values could be worth calculated on g or 100 g for the uniformity of results.

Best regards,

Reviewer.

Author Response

Reviewer 3:

Comment 1: Subsection 2.3.1. Page 5, line 186. The “rpm” should be converted in “g”. Please, recalculate “rpm” in “g”.

Answer 1: According to the suggestion, the authors had recalculated from rpm to g.

Comment 2: Do I understand correctly that the data in Table 4 and Figure 2 duplicate each other? If so, please, leave Table 4 or Figure 2, not both.

Answer 1: As suggested, the authors removed Figure 2 and inserted on Table 4 the statistically significant differences.

Comment 3: Table 6. The other results expressed on g or 100 g, why Total phenolic compound is expressed on L? The analysis could only be performed directly on the aqueous extracts, but a certain sample (g or mg) of algae was used for preparing of aqueous extracts, therefore the values could be worth calculated on g or 100 g for the uniformity of results.

Answer 1: In accordance with the suggestions, the authors had recalculated the total phenolic content to standardize the results.

Round 4

Reviewer 3 Report

Dear Authors,

Thank You for careful revising the manuscript.

Here are my additional comments:

  1. Subsection 2.7. and Table 6.

It should be noted, how the results are expressed. “To quantify the total phenolic content, a standard curve was performed (y =0.0168x 318 + 0.0159; r2=0.9998) with different concentration of Gallic acid (0, 4, 6, 8, 10, 20, 40, 60 mg GAE/ L).” And then should be noted, that, for e.g. the results “results were expressed in mg-equiv. gallic acid /g algae material”. Did fresh or dry matter was used for recalculation of results? Please, check also Table 6, because the units are still unclear.

Best regards,

Reviewer.

Author Response

Reviewer 3:

Comment 1: Subsection 2.7. and Table 6.

It should be noted, how the results are expressed. “To quantify the total phenolic content, a standard curve was performed (y =0.0168x 318 + 0.0159; r2=0.9998) with different concentration of Gallic acid (0, 4, 6, 8, 10, 20, 40, 60 mg GAE/ L).” And then should be noted, that, for e.g. the results “results were expressed in mg-equiv. gallic acid /g algae material”. Did fresh or dry matter was used for recalculation of results? Please, check also Table 6, because the units are still unclear.

Answer 1: The authors would like to acknowledge the reviewer for the insightful comments on the manuscript that helped to improve it.

Regarding the expression of the total phenolic content data, this information is specified on lines 311-314 (subsection 2.8), as previously suggested.

The results are expressed on a dry weight basis, thus in conformity with the remaining data, it is expressed by g GAE/ 100g algae (Dry weight). Moreover, Table 6 was reviewed and corrected.